# Antioxidants as Therapeutic Agents in Acute Respiratory Distress Syndrome (ARDS) Treatment—From Mice to Men

**DOI:** 10.3390/biomedicines10010098

**Published:** 2022-01-04

**Authors:** Andreas von Knethen, Ulrike Heinicke, Volker Laux, Michael J. Parnham, Andrea U. Steinbicker, Kai Zacharowski

**Affiliations:** 1Department of Anaesthesiology, Intensive Care Medicine and Pain Therapy, University Hospital Frankfurt, Theodor-Stern-Kai 7, 60590 Frankfurt, Germany; ulrike.heinicke@kgu.de (U.H.); andrea.steinbicker@kgu.de (A.U.S.); kai.zacharowski@kgu.de (K.Z.); 2Fraunhofer Institute for Translational Medicine and Pharmacology ITMP, Theodor-Stern-Kai 7, 60596 Frankfurt, Germany; volker.laux@itmp.fraunhofer.de (V.L.); mjp@epiendo.com (M.J.P.)

**Keywords:** ARDS, antioxidant, Nrf2, GSH, NADPH oxidase, SOD, electron transfer chain

## Abstract

Acute respiratory distress syndrome (ARDS) is a major cause of patient mortality in intensive care units (ICUs) worldwide. Considering that no causative treatment but only symptomatic care is available, it is obvious that there is a high unmet medical need for a new therapeutic concept. One reason for a missing etiologic therapy strategy is the multifactorial origin of ARDS, which leads to a large heterogeneity of patients. This review summarizes the various kinds of ARDS onset with a special focus on the role of reactive oxygen species (ROS), which are generally linked to ARDS development and progression. Taking a closer look at the data which already have been established in mouse models, this review finally proposes the translation of these results on successful antioxidant use in a personalized approach to the ICU patient as a potential adjuvant to standard ARDS treatment.

## 1. Introduction

Acute respiratory distress syndrome (ARDS) was originally described by Ashbaugh et al. in 1967 [1]. Based on 12 patients with an acute onset of tachypnoea, hypoxemia, and diminished compliance due to various causes, the authors postulated a connection between the disease pattern and involvement of the alveolar surface. Positive end-expiratory pressure (PEEP) improved atelectasis and hypoxemia, whereas corticosteroids were helpful in medicating patients suffering from fat embolism or viral pneumonia. In the absence of a uniform definition of ARDS based on its multiple origins, the American-European Consensus Committee on ARDS was established in 1994. Conferences on ARDS took place in Miami, Florida, United States of America, and Barcelona, Spain. Integrating American and European studies, the ARDS experts provided a new definition of ARDS distinguishing between acute lung injury (ALI) and ARDS as its severe form [2]. An acute onset as well as bilateral infiltrates detected by a frontal chest radiograph and a pulmonary artery wedge pressure of above 18 mm Hg or missing evidence of left atrial hypertension were set as common characteristics. Only oxygenation, estimated by the Horovitz index [3], was different in ALI (PaO_2_/FiO_2_ < 300 mm Hg) compared to ARDS (PaO_2_/FiO_2_ < 200 mm Hg). Because of difficulties in adhering to the definition of the chest radiograph and only moderate accordance with the definitions, novel criteria were required when comparing ARDS clinical criteria with autopsy findings [4]. In 2005, Ferguson et al. developed a new clinical ARDS definition based on the Delphi consensus method [5]. Focusing on ARDS as a putative inclusion criterion or endpoint of clinical trials dealing with ALI, the former clinical definition of ARDS was altered. Defined characteristics were hypoxemia with PaO_2_/FiO_2_ < 200 mm Hg and PEEP ≥ 10 mm Hg, acute onset in <72 h, radiographic abnormalities consisting of a bilateral airspace disease involving ≥ two quadrants on frontal chest radiograph, and non-cardiogenic origin, i.e., without clinical signs of congestive heart failure. Moreover, a reduced lung compliance of <50 mL/cm H_2_O (tranquilized patients were adapted to a tidal volume (Vt) of 8 mL/kg, with an ideal body weight and PEEP ≥ 10) and direct or indirect predisposing factors linked to lung injury were defined [5]. In 2011, Raghavendran and Napolitano discussed the existing ARDS definitions and concluded that a new definition was again mandatory. This definition should include pulmonary vs. nonpulmonary risk factors, PaO_2_/FiO_2_ ratio, and standard ventilator settings (PEEP/MAP) < 200 mm Hg and exclude heart failure determined by cardiac echocardiogram [6]. So, one year later the new so-called Berlin Definition was published [7]. With the Berlin definition, ARDS patients are grouped according to disease burden into three states: mild, moderate, and severe. Shared features are acute onset within 1 week, bilateral shadows determined by radiography or computed tomography (CT), and respiratory failure independent of cardiac failure. Differences were now made regarding oxygenation: “mild” is associated with Horovitz 200 mm Hg < PaO_2_/FiO_2_ ≤ 300 mm Hg with PEEP or continuous positive airway pressure (CPAP) ≥ 5 cm H_2_O that can be delivered noninvasively. “Moderate” is characterized by Horovitz of 100 mm Hg < PaO_2_/FiO_2_ ≤ 200 mm Hg with PEEP ≥ 5 cm H_2_O. Finally, “severe” patients are defined by Horovitz index of PaO_2_/FiO_2_ ≤ 100 mm Hg with PEEP ≥ 5 cm H_2_O [7]. As a conclusion, the ALI term was removed. Originally, the Berlin definition was developed for use in high-income countries, with well-equipped intensive care units (ICUs) only. To adapt this setting for less well equipped ICUs, an appropriate ARDS definition was drawn up with modified criteria [8]. These included a SpO_2_/FiO_2_ ≤ 315 because arterial blood gas diagnostics required to examine the PaO_2_/FiO_2_ ratio are often missing. Since mechanical ventilators are also rare, PEEP is not required. Similarly, the lack of X-ray apparatus was taken into account by allowing ultrasonic testing instead.

To date, there is still no appropriate pharmacological treatment regimen available focusing on disease etiology. Intervention is limited to supportive therapy such as lung-protective mechanical ventilation [9] and possibly extracorporeal membrane oxygenation (ECMO) [10], prone positioning [11], positive end-expiratory pressure (PEEP) [12], recruitment maneuvers (RM) [13], neuromuscular blockers [14], and application of systemic corticosteroids [15]. Although high-end medicine techniques are pursued, mortality is still up to 50–80%; therefore, new approaches should be intensively evaluated. To elucidate putative starting points for an improved care strategy, preclinical studies are necessary to prove the efficacies of the tested interventions. 

## 2. Materials and Methods

### 2.1. Mouse Models of ARDS/ALI

Mouse models of ARDS are widely established because breeding and generation times are short. Humanized models as well as gene knockout and knockin mice are available. Last but not least, mouse maintenance is relatively cheap. To induce ARDS in the murine system, two settings can be chosen (Table 1). Lung injury can be provoked directly by targeting the alveolar epithelium or indirectly by damaging the vascular endothelial cells (for a review see [16]). According to the workshop report published in 2011 by a commission, appointed by the American Thoracic Society (ATS), four ARDS features have been assembled for animal models [17]. Focusing on mouse models, the first of these features is histological determination of tissue injury, i.e., neutrophil immigration into the alveolar or interstitial space, formation of proteinaceous debris, and thickening of the alveolar wall. The second feature is the histological breakdown of the epithelial pulmonary capillary barrier, which can be determined by analyzing the extravascular lung water content, using tracers to follow barrier damage or to show an increase in protein content, especially of high-molecular-weight proteins, in bronchoalveolar lavage fluid (BALF) and an augmented coefficient of microvascular filtration. The third aspect focuses on the inflammatory response, which is characterized by an increased number of neutrophils in BALF, raised expression or activity of lung myeloperoxidase (MPO), and enhanced expression of proinflammatory cytokines in BALF or lung tissue. Finally, the fourth relevant attribute is physiological lung failure, indicated by hypoxemia-accelerated alveolar–arterial oxygen differences including ventilation/perfusion mismatches. Three of these four criteria should be detectable to assure ARDS in the mouse. However, in a mouse model, some ARDS characteristics that are obvious in the human patient, such as formation of hyaline membranes, hemorrhage, atelectasis, enhanced lung lymph flow, or high protein concentration in the lymph, are barely detectable. This must be considered in the selection of a mouse ARDS model. As shown in Table 1, models directly linked to lung damage use intratracheal or intranasal application of lipopolysaccharide (LPS) or bacteria, intratracheal instillation of acids or bleomycin, 100% O_2_ and mechanical ventilation (MV) [18], or pulmonary ischemia/reperfusion injury. Whereas the last-mentioned intervention requires surgery, all the other methods depend on intranasal or intratracheal application, which is easier to achieve. The classically used treatment is LPS inhalation, acting via Toll-like receptor (TLR) 4 signaling, leading to symptoms similar to a bacterial infection, as obvious with accumulation of neutrophils in the lung and induction of proinflammatory cytokines. However, due to the missing infectious pathogen, only aspects of a bacterial infection are represented. To induce active infection, living bacteria (such as *E. coli*, *S. aureus*, or *P. aeruginosa*) are administered via the identical route [19,20]. Intratracheal instillation of HCl resembles the aspiration of gastric contents by ICU patients. The use of intratracheal bleomycin mimics a longer-lasting lung damage, which is useful for examining mechanisms leading to fibrosis. Pulmonary ischemia/reperfusion is a frequently used model because it depicts the processes found in patients requiring a lung transplantation.

Often, ARDS is not directly caused by pulmonary damage, but as a secondary event following a different initial insult. Therefore, ARDS is frequently found in patients suffering from sepsis. In mice, this can be achieved by an intraperitoneal (i.p.) injection of bacteria or of a stool suspension (peritoneal cavity infection/PCI), or, with skilled surgical interventions, by cecal ligation and puncture (CLP), the gold standard of sepsis mouse models, or a similar technique requiring a stent implantation into the colon, which is called colon ascendens stent peritonitis (CASP). All of them result in peritonitis. To induce endotoxemia, it is also possible to inject LPS i.p., with the limitations described above. A disease pattern with high proinflammatory cytokine expression as well as strong activation of the immune system results. Depending on the LPS amount applied, the model can also be lethal. Besides sepsis, a model using i.v. oleic acid provokes fat embolism, which sometimes can be found in patients as secondary to bone fracture. ARDS can also occur after multiple traumata, or sometimes when patients receive multiple transfusions. In this case, it is named TRALI for transfusion-induced acute lung injury [21]. Ischemia/reperfusion injuries can also be produced in nonpulmonary tissues such as gut, kidney, or liver, which might be secondary to ARDS. The i.v. injection of H_2_O_2_ can cause ARDS as well. Especially interesting for the ICU patient are so-called two-hit models. These involve a first insult, e.g., development of sepsis or aspiration of gastric content, demanding the use of artificial ventilation as a second hit. This technique takes over active breathing from the lung. Thus, the method has to be adapted to the physiological respiration variables. This means that the pressure used to inflate the lung has to be strong enough to avoid atelectasis but should not be too great to damage pulmonary tissue by overstretching. Because this requires very sensitive pressure setting, ventilation-induced lung injury (VILI) cannot be completely prevented. Hence, modeling these two-hit situations in mice is of great interest for setting up new treatment strategies.
biomedicines-10-00098-t001_Table 1Table 1Mouse models in ARDS research.Direct Lung Damage [17,22,23]Route of ApplicationARDS-Like AffectsAntioxidant Approaches Already UsedRef.LPS [24,25,26]intranasal/intratrachealinstillationlung accumulation of neutrophils, induction of proinflammatory cytokinesNAC, SAMC[27,28,29]Bacteria [30,31,32]intratracheal instillationlung accumulation of neutrophils, induction of proinflammatory cytokinesCDC[33]HCl [34,35,36]intratracheal instillationneutrophil infiltration, damage of alveolar/vascular barrierapocynin, MitoTempo[37,38]Hyperoxia (HALI) [39,40,41]intratrachealdamage of epithelial cells, neutrophil infiltration AA, BNF, SFN, MnSOD[42,43,44]MV (VILI) [45,46,47,48]intratrachealinflammasome-mediated proinflammatory cytokine expressionNAC, Nrf2^+/+^, Nrf2^−/−^, PIP-2; PC-SOD[49,50,51,52]Bleomycin [53,54,55]intratracheal instillationinvertible fibrosisBRNPs, adelmidrol, EC-SOD[56,57,58]Pulmonary ischemia/reperfusion [59,60,61]surgery; mesenteric artery clamping or hilar ligation and reperfusionneutrophil infiltration, damage of alveolar/vascular barrieririsin[60]Indirect lung damage [17,22,23]



Sepsis (live bacteria, CASP, CLP, CSI) [62,63,64,65,66,67]i.p.¸ peritonitisdamage of alveolar/vascular barrierPC-SOD, SOD mimetic, Prdx6^−/−^[51,68,69]Endotoxemia [70,71,72]i.v. or i.p.damage of alveolar/vascular barrierNAC, EUK-8, CypD[73,74,75]Oleic acid [70,76,77]i.v.mimics fat embolismBAY 60-6583, leptin[76,78]Multiple transfusions (TRALI) [79,80,81,82]i.v.; syngeneic or allogenicacute onset; underlying a 2-hit onset, pulmonary neutrophil sequestration, involvement of MΦMΦ depletion, C3^−/−^, C5^−/−^, C5aR^−/−^[83,84,85]Multiple trauma [86,87,88]externally receivedneutrophil infiltration, complement activationp47phox^−/−^[89]H_2_O_2_ [90,91,92]i.v.increased vascular permeability and fluid retention, edema formationAA, TP[93]Nonpulmonary ischemia/reperfusion [94,95,96]surgery; liver, gut, kidneyneutrophil sequestration, acceleration of microvascular permeabilityCypD^Plt^^−/−^, SB239063, FK866, LY333531 [97,98]Two-hit models 



LPS + MV [99,100,101]intratracheal, i.v., i.p.inflammasome-dependentATF3 OE/KD; HIF1α^−/−^, enoxaparin, DJ-1, paracoxib [102,103,104,105]Sepsis + MV [106,107,108]i.p., peritonitis, intratrachealaugment sepsis-mediated organ damageAM [109]HCl + MV [100,110,111,112]intratrachealenhanced HCl impactIL-6^−/−^[113]^+/+^, wild-type mice; ^−/−^, knockout mice; AA, ascorbic acid; AM, adrenomedullin; ATF3, activating transcription factor 3; BAY 60-6583, adenosine A2B receptor agonist; BNF, β-naphthoflavone; BRNPs, bilirubin-derived nanoparticles; C, complement; CASP, colon ascendens stent peritonitis; CDC, water-soluble curcumin formulation; CLP, cecal ligation and puncture; CSI, cecal slurry injection; CybD, cyclophilin D; DJ-1, Daisuke-Junko protein 1; EC-SOD, extracellular SOD; R, receptor; FK866, competitive visfatin inhibitor; HALI, hyperoxia-induced lung injury; HIF, hypoxia-inducible factor; IL, interleukin; i.p., intraperitoneal; i.v., intravenously; KD, knock-down; LPS, lipopolysaccharide; LY333531, PKCβ inhibitor; MΦ, macrophage; MV, mechanical ventilation; NAC, N-acetylcysteine; OE, overexpression; PCI, peritoneal cavity infection; PC-SOD, lecithinized SOD; PIP-2, peroxiredoxin 6 inhibitor peptide-2; PLT^−/−^, platelet-conditional knockout mice; SAMC, S-allylmercaptocysteine; SB239063, p38 MAPK inhibitor; SFN, sulforaphane; TP, α-tocopherol; TRALI, transfusion-induced acute lung injury; VILI, ventilator-induced lung injury.


### 2.2. Oxidative Stress in Pathogenesis of ARDS/ALI

Oxidative stress can be divided into two separate categories. When the intracellular hydrogen peroxide (H_2_O_2_) concentration is lower than 100 nM, this is then called “oxidative eustress”, which is a physiological process important for proliferation, differentiation, migration, and angiogenesis (for a review see [114]). This is opposed to an intracellular reactive oxygen species (ROS) level above 100 nM and up to 10 µM, which is pathological or related to host defense and named “oxidative distress”. It is already well established that the generation of oxidative distress is causative in the pathogenesis and progression of ARDS (for recent reviews see [115,116,117]). In brief, ROS can be formed by dying cells, i.e., apoptotic or necrotic cell death [118], as a product of the mitochondrial respiratory chain (Figure 1a) [119] or to fight infections by cells of the innate immune system, such as neutrophil granulocytes or macrophages, by activation of the NADPH oxidase, which in this case is the Nox2 (Figure 1b) [120]. In the lung, accordingly, excessive ROS are formed by damaged pulmonary endothelial and epithelial cells as well as by infiltrating leukocytes, which are predominantly neutrophils [121]. Moreover, alveolar and airway epithelial cells as well as vascular endothelial cells express Nox4 (Figure 1c) [122,123]. This NADPH oxidase, in contrast to Nox2, generates H_2_O_2_ and not O_2_^−^ [128]. Although Nox4-produced H_2_O_2_ is an important second messenger, when its expression is induced or its function is activated, it also contributes to lung injury [124].

Altering the redox balance by increasing the amount of generated ROS is counterbalanced in part by detoxifying enzymes such as superoxide dismutases (Figure 2). Three isoforms of this enzyme family exist: SOD1 (CuZnSOD), SOD2 (MnSOD), and SOD3 (EC-SOD). These enzymes catalyze the transition from superoxide O_2_^−^ to H_2_O_2_. Although all three enzymes catalyze the same reaction, the proteins differ in their cellular localization [131]. SOD1 is located in the cytoplasm and is important for the removal of O_2_^−^ mainly derived from NADPH oxidases [132]. SOD2 is predominantly found in mitochondria and is responsible for the conversion of O_2_^−^ generated from oxidative phosphorylation reactions [133]. Finally, SOD3 is an extracellular enzyme found in the blood and attached to the extracellular matrix, mainly expressed in the lung. SOD3 is important to reduce pulmonary ROS [134,135]. H_2_O_2_ can be further metabolized to the hydroxyl radical by the Fenton reaction (Fe^2+^ + H_2_O_2_ → Fe^3+^ + ˙OH + OH^−^), the second part of the Haber–Weiss mechanism, leading to the hydroxyl radical (˙OH), which is highly antimicrobial and injurious to cells [136]. To prevent cellular damage, especially in macrophages, it can be completely detoxified by catalase to H_2_O and O_2_ (Figure 2b) [137]. In neutrophils, it can be converted by myeloperoxidase (MPO), in the presence of a chloride anion (Cl^−^), to hypochlorous acid (HOCl^−^) [138]. Besides enzymes, the redox-sensitive tripeptide γ-L-glutamyl-L-cysteinyl-glycine (glutathione/GSH) is also an important cellular antioxidant [139]. However, the capacity of these naturally occurring antioxidants is limited [140]. This explains why excessive ROS formation cannot be adequately compensated, consequently leading to cell and organ damage [141,142,143]. In ARDS, the breakdown of the endothelial/epithelial barrier is an important hallmark of disease progression [144,145]. Dysfunction of this barrier in ARDS (depicted in Figure 3) is associated with the release of danger-associated molecular patterns (DAMPs), such as HMGB1, from damaged cells [146] or pathogen-associated molecular patterns (PAMPs) such as lipopolysaccharides (LPS), originating from the outer membrane of Gram-negative bacteria, or lipoteichoic acid (LTA), arising from Gram-positive bacteria acting via Toll-like receptor-4 or -2 dependent signaling (Figure 3b) [147,148,149]. Originally located in the alveoli, these DAMPs/PAMPs can now diffuse into the vasculature, leading to systemic inflammation (Figure 3c–f). In return, neutrophil granulocytes can more easily pass the endothelial layer when recruited from the blood vessels into the alveoli (Figure 3b) [121]. Thus, as shown in Figure 3c,d, excessive fluids can provoke pulmonary edema, which accumulates in the lung interstitium, inhibiting gas exchange, which is closely associated with hypoxemia, requiring artificial respiration. When edema clearance is appropriately initiated, resolution of the inflammation phase follows (Figure 3e), characterized by phagocytosis of apoptotic neutrophils by MΦ and proliferation or differentiation of alveolar epithelial cells (AT-I and -II), and the function of alveoli is restored (Figure 3f).

Mechanistically, ROS contribute to the damage of the epithelial barrier. ROS-dependent induction of the matrix metalloproteinase (MMP)-9 causes damage, internalization, and downregulation of proteins of intercellular connections, so-called tight junctions [150], such as claudins, occludins, and E-cadherins, linking the extracellular glycocalyx with the intracellular cytoskeleton (Figure 4a) [151,152]. The loss of cell–cell interactions consequently is associated with an increase in permeability and gap formation [117], leading to edema formation [153]. This is promoted by the similarly reduced expression of the epithelial sodium channel (ENaC), responsible for fluid retention, thus generally counteracting the development of edema [122]. ROS-mediated modification of cysteine 43 of the β-subunit of ENaC permits its ubiquitination by the E3 ubiquitin-protein ligase NEDD4-2, leading to its proteasomal or lysosomal degradation (Figure 4b) [154]. In line with this, a conditional knockout of NEDD4-2 in lung epithelial cells supports this observation [155]. However, the knockout has an adverse effect by lowering the epithelial humidity, favoring fibrosis [155]. Increased ROS, as observed in lungs of PKCα-knockout mice, enhanced ENaC internalization and reduced ENaC expression [156]. The ROS scavenger tempol reversed this effect, further supporting a role of ROS in ENaC regulation [156].

In addition to the effect of ROS on cell–cell interactions, ROS are also involved in altering coagulation and fibrinolysis, contributing to ARDS (Figure 5) [159,160,161]. However, further research is necessary to understand the underlying principles in detail [162]. Studies already performed have shown that ROS mainly contribute to a proinflammatory response, which is associated with activation of platelets and subsequent trapping in the microcirculation leading to thrombocytopenia [163,164]. Nox2 activation is critically involved in platelet activation and thrombosis in response to oxidative stress [165]. The coagulation factors XIIa (FXIIa) and tissue factor (TF) are involved in ROS-mediated effects as well. The contact factor FXIIa, activated by polyphenols released from activated platelets, is an important component in clot formation [166,167], and TF promotes neutrophil extracellular trap (NET) formation, a process which is called NETosis [168], and disease progression [169]. NETosis is directly associated with ROS release, as shown in mice where NADPH oxidase inhibition blocked NETosis and improved thrombosis [163]. Complement activation, a characteristic event in infection, is involved in thrombosis and disseminated intravascular coagulation (DIC). This is especially true when complement activation is unbalanced [170]. The complement factor C5a is an established chemotactic factor attracting immune cells such as neutrophils and macrophages. Subsequently, by binding to the phagocytes, C5a activates a G-protein-coupled signal pathway, leading to Nox2-dependent ROS formation [171]. A similar ROS effect was observed in murine kidney endothelial cells, leading to mitochondria-dependent apoptosis in response to C5a [172]. In addition to compounds leading to coagulation, endogenous fibrinolysis is also reduced. Although fibrinolysis is initially activated, this is counteracted by the simultaneous and sustained upregulation of the plasminogen activator inhibitor-1 (PAI), resulting in enhanced coagulation [173]. Taken together, these mechanisms, in combination, ensure coagulation, thus ensuring that the risk of DIC occurrence remains high [174].

### 2.3. Antioxidative Treatments

In view of the role of ROS in ARDS development and progression, various possibilities exist to prevent their damaging impact. Pharmacologically, effector proteins, i.e., enzymes involved in ROS formation, can be inhibited, factors important in ROS detoxification can be activated, or their function mimicked. First, we focus on compounds that can be used pharmacologically to reduce ROS formation.

#### 2.3.1. Pharmacological Antioxidants

Pharmacologically, differentiation should be made between factors inhibiting ROS-generating enzymes such as apocynin, developed to inhibit Nox2, thus preventing ARDS in a CLP mouse model [176], or setanaxib, also known as GKT 137831, to block Nox4-dependent H_2_O_2_ generation, which in turn reduced lung ischemia/reperfusion injury (LIRI) in mice [177], together with so-called ROS scavengers. These are crucial to detoxify ROS. Among them, classical SOD mimetics such as tempol [178] and EUK-8, which is also a catalase mimetic [73], or SOD-derived compounds such as lecithinized SOD2 (PC-SOD) [51], or recombinant SOD1 [179] and the mitochondria-targeted antioxidants MitoQ, MitoTempo, or tiron are available [180]. The mitochondria-targeted antioxidants are especially suitable when ROS are generated by the mitochondrial ETC and were found to improve VILI in mouse preclinical settings [181]. However, the most important endogenous compound in terms of antioxidant capacity is GSH [182]. This is also true in lung inflammation [183]. As shown in Figure 2b, it is indispensable to the function of glutathione peroxidase (GPx), which detoxifies H_2_O_2_ to H_2_O by oxidizing two GSH to one GSSG, which is accordingly the oxidized form of GSH [139]. Then, the GSH pool is restored by glutathione reductase (GR), which reduces GSSG back to GSH. Based on the pharmacologically available GSH precursor N-acetylcysteine (NAC) (Figure 6), the quantity of GSH can be adjusted therapeutically. This has been investigated also in infectious diseases, where NAC improved the disease pattern and survival in the murine system [49,73,184]. Interestingly, contradictory results have also been observed [185,186]. Therefore, the single use of NAC does not appear to be a sufficient treatment regime in the mouse model [187]. The organoselenium compound ebselen, under development for therapy of a variety of clinical conditions involving oxidative stress, reacts with GSH in a cyclic mechanism similar to that of GPx, thereby inactivating hydroperoxides, including H_2_O_2_ and regenerating GSH [188,189,190]. Oral administration of ebselen inhibits ozone-induced lung inflammation in rats [191]. It has also been proposed as a potential treatment for respiratory inflammation in COVID-19 infection, since it also reacts with the free thiol group of the main protease (Mpro) of the coronavirus SARS-CoV-2 to inhibit the protease [192].

Vitamin C has also been identified as an antioxidant with a protective function in murine acute lung injury [193]. The mice received either ascorbic acid (AscA) or dehydroascorbic acid (DHA) 30 min following abdominal sepsis induction by intraperitoneal cecal slurry injection [194]. Both treatments reduced lung injury by differing mechanisms, involving reduced epithelial barrier breakdown, maintaining alveolar fluid clearance, preventing tight junction loss, and inhibiting rearrangement of the cytoskeleton [194].

#### 2.3.2. Nrf2

The most important transcription factor in regulating antioxidant genes is the cap-and-collar type transcription factor, nuclear factor erythroid 2-related factor 2 (Nrf2) [195]. Its expression is regulated under healthy conditions by Keap1, which targets Nrf2 for proteasomal degradation by the Cullin3/Rbx1 ubiquitination system [196]. Upon (electrophilic) oxidative stress, cysteines in Keap1 are oxidized and Nrf2 is released [196], consequently leading to its stabilization and subsequent induction of target gene expression [197]. In keeping with its role as a transcription factor, Nrf2 activates the expression of factors involved in detoxifying ROS, such as glutamate-cysteine ligase catalytic subunit (GCLC), glutamate-cysteine ligase modifier subunit (GCLM), GPx, GR, heme-oxygenase (HO-1), Prx, and SOD3 [197]. Thus, activation of Nrf2 leads to an antioxidative response, which is generally associated with an improved outcome of lung injury in mouse models [198]. In line with this, Nrf2-deficient mice show enhanced acute lung injury compared to wild-type mice following intestinal ischemia/reperfusion [199]. Considering that Nrf2 binds to antioxidative-response elements (AREs) in the promoter regions of target genes, their activation can also be followed in an ARE-reporter mouse, where the ARE site has been associated with a luciferase (Luc) reporter gene. In these mice, Nrf2 activation provokes luciferase expression in parallel, which can be determined by bioluminescence imaging [200]. In association with its anti-inflammatory function, Nrf2 prevents classical activation of MΦ to an M1 phenotype and promotes alternative activation, resulting in an M2 MΦ phenotype. This was reversed by an Nrf2 siRNA approach knocking down the transcription factor [201]. In these studies, established pharmacological treatments, such as tert-butylhydroquinone (tBHQ) [201], bardoxolone (CDDO) [202], resveratrol [203], and sulforaphane (SFN) [200], stabilized Nrf2 by modulating its binding to Keap1. Interestingly, stabilization of Nrf2 has also been shown to occur indirectly. Among others, the tyrosine kinase inhibitor dasatinib counteracts LPS-dependent ALI. This effect was associated with increased Nrf2 expression and activation as observed by an enhanced expression of the Nrf2 target gene HO-1 in lung tissue [204]. This treatment promotes a similar M2 polarization of MΦ. The above-mentioned mitochondria-targeted ROS scavenger mitoQ reduced ROS formation in an intraperitoneal LPS mouse model by activating Nrf2 and the corresponding expression of target genes [205]. This improvement was largely abolished in Nrf2 knockout mice. Mechanistically, mitoQ-mediated Nrf2 stabilization induced the expression of HO-1, which in turn blocks LPS-upregulated expression of the mitochondria fission factor Drp1 in alveolar epithelial cells [206]. Thus, fission of mitochondria, associated with cytochrome c release, concomitant caspase 3 activation, and apoptosis leading to the breakdown of the epithelial barrier, is prevented [206]. Since sex-specific effects in the response to antioxidative treatments are possible, Callaway et al. [42] demonstrated in a hyperoxic lung injury model that Nrf2 knockout especially in female animals reduced survival. Independently of sex, treatment with the cytochrome P450 (CYP) 1A inducer β-napthoflavone (BNF) improved animal survival, eliminating sex differences [42]. Therefore, these data might open up a novel option for a treatment regime in patients with a relative NRF2 deficiency. Furthermore, it has been shown that Nrf2 induction is associated with a decrease in NF-κB activation [207]. However, recent data suggest that Nrf2 is activated in response to mild hyperoxia (30% O_2_), whereas following high hyperoxia (100% O_2_) the increase in oxidative stress is linked to the activation of Nrf2 and NF-κB in parallel. Very high hyperoxia (140% O_2_) activated NF-κB almost exclusively [208].

#### 2.3.3. PPARγ

This ligand-dependent nuclear hormone receptor PPARγ was identified on the basis of its role in glucose metabolism. Recent studies support a second role of PPARγ as an anti-inflammatory, sometimes proapoptotic factor. It has been shown that activation of PPARγ inhibited activation of Nox2 in macrophages [209]. It is an important regulator of MΦ polarization, which was further substantiated using PPARγ knockout mice [210,211]. Moreover, PPARγ is also a target of direct redox regulation, i.e., cysteine-residue modification [212]. In ARDS research, activation of PPARγ with the ligand rosiglitazone restored surface expression of the ENaC channel on alveolar type II epithelial cells (Figure 3a) and ameliorated acute lung injury in an intratracheal LPS mouse model, showing upregulation of ENaC expression as well [213]. In the study by Wang et al. [214] activation of PPARγ attenuated LPS-induced acute lung injury by preventing HMGB1 release and decreasing RAGE levels, both known to be upregulated in ARDS mouse models [215]. Considering PPARγ as a negative regulator of macrophage activation [216], metabolic and epigenetic changes leading to PPARγ expression or activation can also alter the phenotype of alveolar MΦs, which are important contributors to sepsis-initiated ARDS. In this regard, α-ketoglutarate, originating as an intermediate from the tricarboxylic acid (TCA) cycle, has recently been characterized as a PPARγ agonist [217]. This is especially important in classically activated, so-called M1 MΦ, known to show a reduced TCA circle with an emphasis on the glycolytic metabolism to generate ATP, compared to alternatively activated M2 MΦ, relying on oxidative metabolism to produce ATP [218,219]. Thus, Liu et al. observed an α-ketoglutarate-dependent inhibition of alveolar MΦs, which was corroborated by the cellular localization of PPARγ, which was mainly nuclear after α-ketoglutarate exposure in the murine alveolar MΦ cell line MH-S [220]. In connection with the epigenetic regulation of PPARγ, Bao et al. demonstrated that a key epigenetic regulator, the histone methyltransferase enhancer of zeste homolog 2 (EZH2), alters the expression profile of alveolar MΦ in LPS-mediated ARDS in mice [185]. Genetic knockdown of EZH2 and pharmacological inhibition of EZH2 by 3-dezaneplanocin suppressed an M1 MΦ phenotype, while promoting M2 MΦs, by permitting activation of STAT6 and PPARγ [185]. Correspondingly, inhibition of PPARγ is associated with an enhanced inflammatory immune response, for instance with fibroproliferative ARDS following bleomycin intratracheal instillation [221]. However, endogenously activated PPARγ also provoked T cell depletion in a CLP mouse model, contributing to immune paralysis and a worse outcome [222]. Such interventions in sepsis could, therefore, only be considered with a close assessment of the ongoing immune status.

### 2.4. Mouse Data Translated to the ARDS Patients’ Situation

Although antioxidative therapy concepts have been successfully employed in the mouse model as outlined above, translating these approaches to the human patient has not been very successful. One reason for this failure is the heterogeneous multifactorial etiology of ARDS [223,224], which makes it difficult to establish an effective standard treatment regime. Moreover, in the mouse models, mainly young mice of one sex have been used to determine whether the individual treatments improved animal survival. Generally, in animals, this type of study does not require any IC support during the experimental phase, which is of course mandatory in the human ICU. Therefore, disease progression in the mouse model cannot be adequately followed. Experimental settings are planned to control the health state of mice, providing estimates of the time of disease onset, which is crucial to start therapy on time and to draw firm conclusions from the data obtained. However, this rating is vague, and accordingly, the evidence generated is limited. Hence, it would be more appropriate to follow disease progression in the animal with similar equipment to that in the human ICU which facilitates adequate translation of the mouse data to the human situation. In addition, the mouse metabolic rate is significantly different from humans [225], which is even enhanced during sepsis [226]. Moreover, mouse models need to be extended to experiments with aged mice, thus better reflecting the ICU situation. Keeping in mind the different conditions that ultimately result in ARDS, also in the murine system [100], it would be appropriate to group data from the animals according to the specific etiologies, courses of disease, treatments, and subsequent responses. Correspondingly, grouped animals might lead to the identification of different treatment regimens (Figure 7a). Broadening the marker panel would possibly allow the inclusion of slightly differing animals into the same test groups, to determine whether these also respond to the treatment (Figure 7b). Finally, with treatment concepts established in this preclinical setting, data from clinical trials can be reanalyzed in relation to the mouse results, allowing similar group assignments (Figure 7c). Transferring these retrospective analyses to the active ICU may open new therapeutic concepts with a special focus on personalized medicine. First attempts have been proposed and already been made to subphenotype ARDS patients [227,228,229,230], although some data are discouraging [231].

## 3. Conclusions and Outlook

Recently, two reviews summarized pharmacological treatments for patients suffering from ARDS [232,233]. Although, the main message of both reviews is that there is no appropriate pharmacological therapy option, but only supportive care, they agree on the necessity to subgroup ARDS patients to identify personalized treatment concepts. With respect to examples of mouse models of ARDS discussed here, which respond to antioxidative treatment, putative care concepts in the ICU might also include an antioxidative approach as a component of the medical care. This seems especially interesting since, as an antioxidative concept concerning viral infections associated with ARDS described here, the use of NAC as a putative (adjuvant) therapy concept in COVID-19 infection, like that of ebselen mentioned previously, has also been suggested [234] and summarized in [235].

## Figures and Tables

**Figure 1 biomedicines-10-00098-f001:**
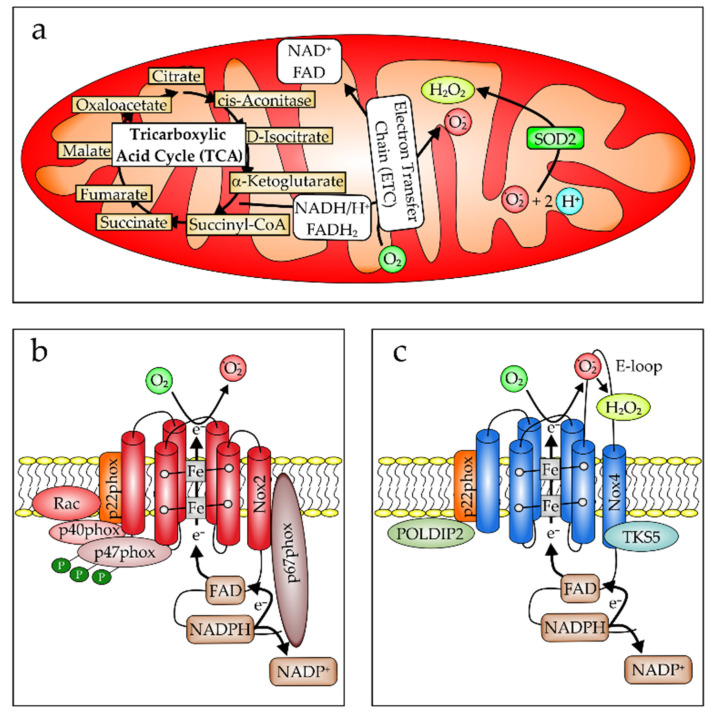
Intracellular ROS production. (**a**) Mitochondria are an important source of intracellular ROS production (mod. from [125]). Being responsible for cellular ATP generation, mitochondria contain the electron transport chain (ETC), and when uncoupled or damaged, ROS can be formed accidentally. The ETC is located in the inner mitochondrial membrane. However, complexes I and III of the respiratory chain also mainly produce O_2_^−^ in intact mitochondria [126], contributing to the cellular redox load [127]. (**b**) Phagocytes such as neutrophils and monocytes/MΦ express Nox2. This is a component of a multiprotein complex, formed in the cell membrane upon cell activation. Besides Nox2, which is also named gp91phox, the subunits p40phox, p47phox, p22phox, and p67phox are required to transfer an electron from NADPH to FAD and then via the Fe of the two associated heme groups to O_2_, leading to generation of the superoxide radical O_2_^−^. (**c**) In contrast, Nox4 is expressed mainly in endothelial and epithelial cells, where it is located at the endoplasmic reticulum and mitochondria. Nox4 only requires the additional subunit p22phox for ROS production, which, in contrast to Nox2, is situated on the E-loop and associated with a direct dismutation of O_2_^−^ to O_2_ and H_2_O_2_ (mod. from [128]). Because it is constitutively active, Nox4 is regulated by its expression and by binding to factors such as Poldip2 and tyrosine kinase substrate with five SH3 domains (TKS5) (mod. from [129,130]).

**Figure 2 biomedicines-10-00098-f002:**
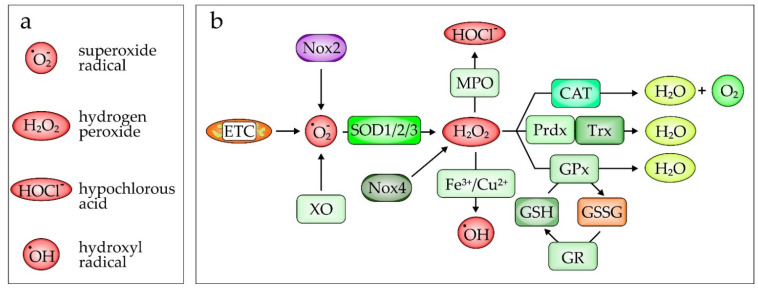
Reactive oxygen species (ROS) and ROS-generating and -scavenging enzymes. (**a**) ROS involved in ARDS. (**b**) The superoxide radical O_2_^−^ is generated by the NADPH oxidase 2 (Nox2), the xanthine oxidase (XO), and the electron transfer chain (ETC) located in the mitochondria. O_2_^−^ is dismutated to hydrogen peroxide (H_2_O_2_) by one of three superoxide dismutases (SODs), which are located in the cytosol (SOD1 ≙ CuZnSOD), in mitochondria (SOD2 ≙ MnSOD), or extracellularly, often associated with the extracellular matrix (SOD3 ≙ EC-SOD). One further source of H_2_O_2_ is Nox4, which is located in mitochondria or endoplasmic reticulum (ER) of endothelial as well as epithelial cells. H_2_O_2_ is the substrate for the myeloperoxidase-(MPO)-derived oxidant hypochlorous acid (HOCl^−^), known to cause tissue injury. Stored in neutrophil granules, MPO is released following neutrophil activation. In the Fenton reaction, H_2_O_2_ is further metabolized to the highly antimicrobial hydroxyl radical (˙OH). ROS-scavenging enzymes, such as catalase (CAT) or glutathione peroxidase (GPx), detoxify H_2_O_2_ to H_2_O and O_2_. To achieve this, GPx oxidizes GSH to GSSG, which in return is reduced via the glutathione reductase to GSH. Similarly, peroxiredoxin (Prx), belonging to a small family of peroxidases, reduces H_2_O_2_ by oxidizing thioredoxin (Trx), which then is restored to the reduced form by the thioredoxin reductase (not shown).

**Figure 3 biomedicines-10-00098-f003:**
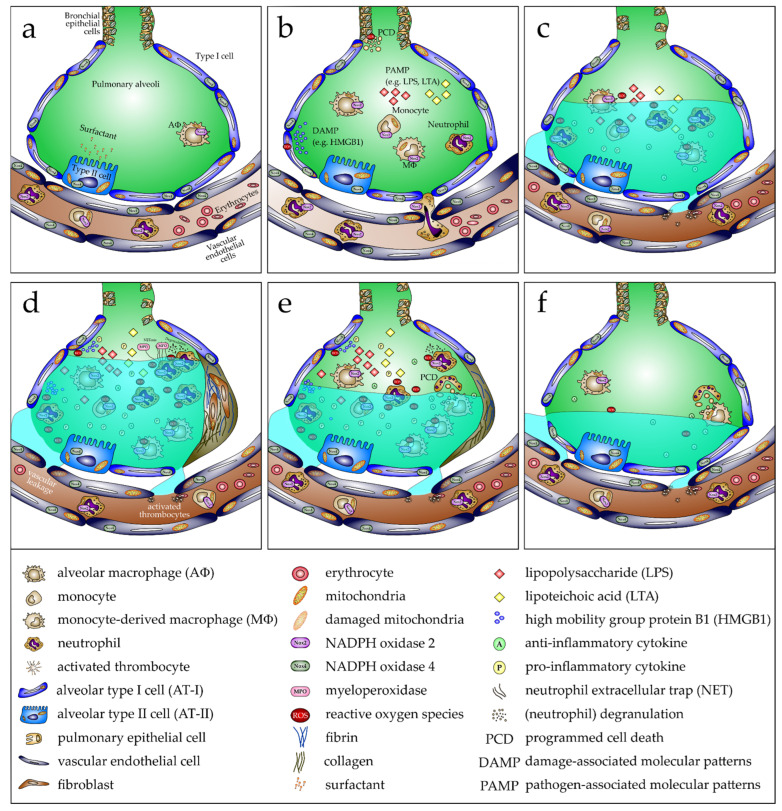
Development, progression, and resolution of ARDS in response to pathogen-associated molecular patterns (PAMPs). (**a**) In the healthy lung, alveoli show no neutrophil infiltration and only limited alveolar macrophages (AΦ). Alveolar type II cells (AT-II) produce adequate surfactant to keep the alveolar epithelium covered effectively, thus reducing the surface tension, necessary to prevent a collapse of the alveoli after expiration. Consequently, optimal gas exchange occurs. (**b**) Following inhalation of bacteria or bacterial components such as lipopolysaccharide (LPS) or lipoteichoic acid (LTA), which are PAMPs, bronchial and alveolar epithelial cells are activated, increasing expression of proinflammatory chemokines and cytokines. This provokes infiltration of immune cells, mainly neutrophils and some monocytes, from the bloodstream. Consequently, the proinflammatory response is enhanced, including proinflammatory cytokines and mediators such as reactive oxygen species (ROS), produced primarily by the phagocytic NADPH oxidase (Nox2) expressed by neutrophils and monocytes/MΦ. Programmed cell death (PCD) of bronchial epithelial cells is induced and HMGB1 as a damage-associated molecular pattern (DAMP) from alveolar type I cells (AT-I) is released, which is also associated with cell demise. (**c**) Cell death is linked to the damage of the alveolar–capillary barrier, causing lung edema, which significantly reduces lung function with reduced blood gas exchange. (**d**) High numbers of neutrophils and MΦ in the alveoli, a consequence of cell death and a proinflammatory environment, facilitate proliferation of fibroblasts, expressing fibronectin and collagen. These contribute to fibrosis and reduce the normal function of the alveoli. (**e**) Reduction of the proinflammatory profile emphasizes an anti-inflammatory response and fibrosis is reversed. Proliferation of alveolar and bronchial cells closes the gap, which arises due to prior cell death. Therefore, lung edema abates. Finally, (**f**) fibrosis completely reverses, and the cell composition of alveoli is almost completely restored. When edema is also entirely resolved, lung function is reestablished.

**Figure 4 biomedicines-10-00098-f004:**
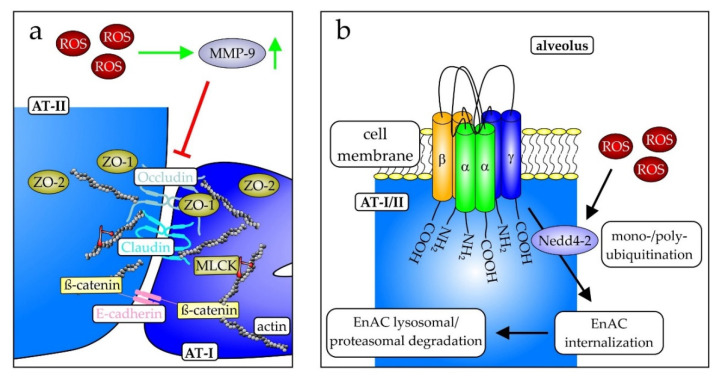
Important structures in alveolar epithelial cells. (**a**) The connection between alveolar type I and type II cells is mediated by occludin and claudin, two proteins involved in the formation of tight junctions, and the calcium-dependent cell adhesion protein epithelial (E)-cadherin. These proteins connect cells to the intracellular actin filaments and downstream signaling cascades as exemplified by the myosin light chain kinase (MLCK) via β-catenin and the zona occludens proteins ZO-1 and ZO-2 (mod. from [117]). (**b**) The epithelial sodium channel (ENaC) is a multimeric protein complex localized in the cell membrane of the pulmonary AT-I and -II cells. It consists of the three homo-dimeric subunits αα, ββ, and γγ and is an important mediator of pulmonary edema clearance and is expressed in two isoforms. One is highly Na^+^ selective, whereas the other is a cation-nonselective form. During ARDS, several mechanisms provoke downregulation of ENaC expression, apical localization, and activity. ENaC is downregulated by internalization and proteasomal or lysosomal degradation, following Nedd4-2-dependent mono- or poly-ubiquitination. During infection, the downregulation of inflammatory cytokines TNF-α, TGF-β, IL-4, IL-13, and IL-1β contributes to this [mod. from [157,158]).

**Figure 5 biomedicines-10-00098-f005:**
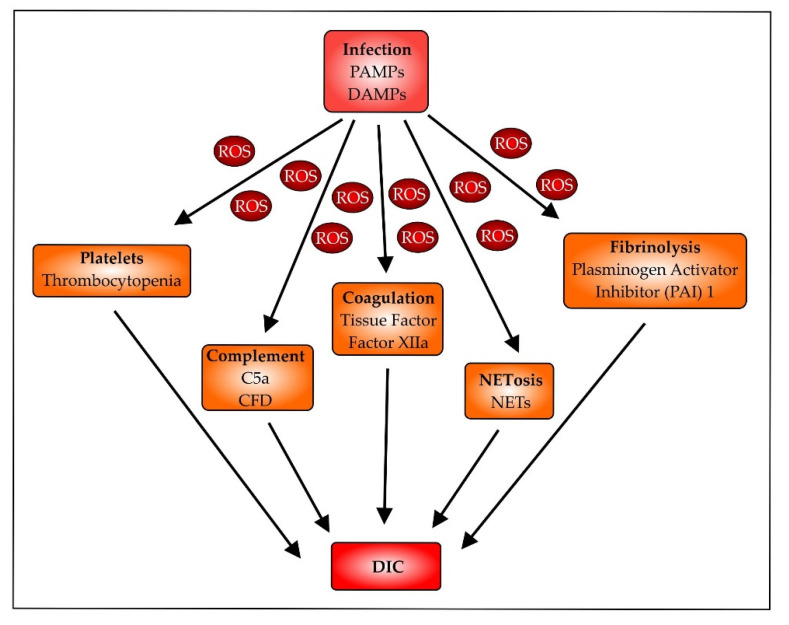
Disseminated intravascular coagulation (DIC). Infectious pathogens via their associated PAMPs release resultant DAMPs, following cell activation or damage, leading to ROS production. These proinflammatory mediators contribute to the activation of platelets, leading to thrombocytopenia [175]; attract immune cells due to the liberation of the chemotactic complement factor C5a [89]; induce tissue factor (TF) production by endothelial cells [169]; and increase coagulation factor FXIIa [166] and plasminogen activator inhibitor (PAI) I, reducing fibrinolysis [173].

**Figure 6 biomedicines-10-00098-f006:**
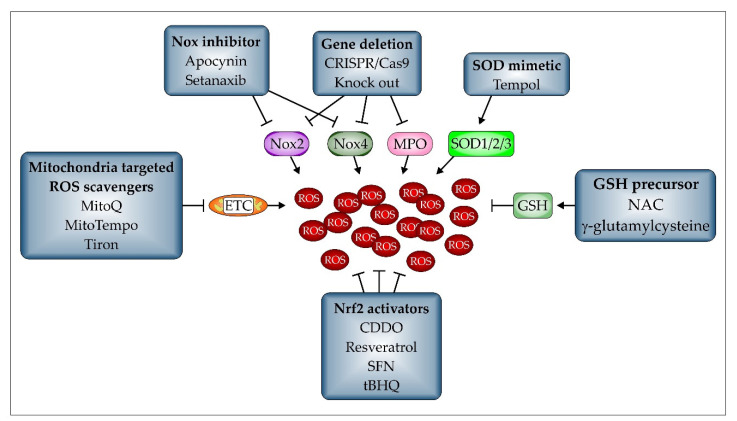
Therapeutic concepts to reduce and prevent ROS formation. Several approaches have been tested in the murine model to reduce and prevent ROS formation during lung inflammation. Considering the major factors involved in ROS formation, such as Nox2, Nox4, MPO, and the ETC, specific inhibitors or compounds targeted to mitochondria were shown to be effective in improving ALI. SOD mimetics, which only reduce the amount of generated O_2_^−^, have also been found to have an impact. Additionally, GSH precursors, maintaining a high intra- and extracellular GSH-pool, leading to a more reductive environment, are potent in depleting ROS. Finally, activators leading to the stabilization and thus activation of the transcription factor Nrf2 have been shown to significantly contribute to the expression of factors important in detoxifying ROS. However, one further possibility in the murine system, gene deletion, is still difficult to achieve in the human patient.

**Figure 7 biomedicines-10-00098-f007:**
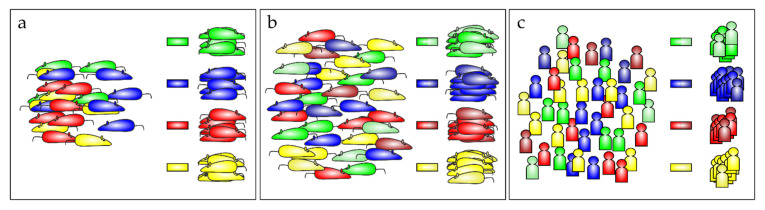
Precision medicine from mice to men. (**a**) Grouping animals according to the characteristics of ARDS origin or disease pattern might be advantageous for the optimization of treatment. However, (**b**) whether a more global classification, integrating some broader aspects, will provide a greater advance, needs to be tested. (**c**) This approach is also likely to be helpful for patients suffering from ARDS, allowing specific personalized treatment of the corresponding patient group.

## Data Availability

Not applicable.

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
