# Peer review of "Antioxidants as Therapeutic Agents in Acute Respiratory Distress Syndrome (ARDS) Treatment—From Mice to Men"

_biomedicines, 2022, doi:10.3390/biomedicines10010098_

Round 1

Reviewer 1 Report

The authors present an interesting review on the antioxidant’s agents in ARDS. The authors must be commended for the exhaustive review and the pedagogic quality of the writing, without going into unnecessary detail.

I really appreciate the very good description and explanations on the complex pathways involved in the proposed review.

I just have two small remarks:

There is no mention of the NF Kappa B interplay with NRF2 which I believe is of interest in the ARDS mechanism, recent articles are discussing such interplay and are referencing some fundamental results, for instance the following (among others) : Fratantonio D, Virgili F, Zucchi A, Lambrechts K, Latronico T, Lafere P, Germonpre P & Balestra C. (2021). Increasing Oxygen Partial Pressures Induce a Distinct Transcriptional Response in Human PBMC: A Pilot Study on the "Normobaric Oxygen Paradox". Int J Mol Sci 22.

The other remark would be on the paragraph 2.4 regarding the mouse model. We agree on the fact that transition from mice to humans isnot always easy, and particularily when oxidation is in discussion, in fact metabolic rate is clearly different compared to humans it may be raised if the authors are willing to.

Thank you for letting me review such an interesting manuscript.

Reviewer 2 Report

Authors explored the possibility of antioxidative therapeutic modality in the treatment of patients suffering from ARDS. Despite the large heterogeneity of ARDS and its multifactorial origins, careful analysis of patients and grouping might shed new light on improved (personalized and precise) treatment of ARDS patients in near future.

Although this maybe a timely and much needed review on ARDS in mouse models and clinical translation of antioxidants in humans, editing by a speaker of native English is recommended. The following typos or errors must be addressed.

Page 2, line 9: PaO2/FiO2

Page 2, line 19 :  H20

Page 2, line 26:  x-ray -> X-ray

Page 6, line 3 from bottom: SOD1 (CuZnSOD), 2 (MnSOD), and 3 (EC-SOD), -> SOD1 (CuZnSOD), SOD2 (MnSOD), and SOD3 (EC-SOD),

Page 9, Figure 4 legend: the downregulation of inflammatory cytokines TNF-, TGF-, IL-4, IL-13, and IL-1 contribute to this. -> ~ contributes to this.

Page 11, line 4 from bottom: '30 min following abdominal sepsis induction~' should be rewritten because the sentence should not start with numbers.

Page 12, line 15: Nrf2-defifcient -> Nrf2-deficient

Page 12, line 16 from bottom: Drp1in -> Drp1 in
